# Development of Energy-Efficient Superhydrophobic Polypropylene Fabric by Oxygen Plasma Etching and Thermal Aging

**DOI:** 10.3390/polym12112756

**Published:** 2020-11-23

**Authors:** Shinyoung Kim, Ji-Hyun Oh, Chung Hee Park

**Affiliations:** 1Department of Textiles, Merchandising and Fashion Design, Seoul National University, Seoul 08826, Korea; annakim95@snu.ac.kr (S.K.); joh25@ncsu.edu (J.-H.O.); 2Department of Chemical and Biomolecular Engineering, North Carolina State University, Raleigh, NC 27695, USA

**Keywords:** polypropylene, plasma etching, thermal aging, superhydrophobicity, weave density, self-cleaning, energy-efficient

## Abstract

This study developed a human-friendly energy-efficient superhydrophobic polypropylene (PP) fabric by oxygen plasma etching and short-term thermal aging without additional chemicals. The effect of the microroughness on the superhydrophobicity was examined by adjusting the weave density. After the PP fabric was treated with oxygen plasma etching for 15 min and thermal aging at 120 °C for 1 h (E15H120 1 h), the static contact and shedding angles were 162.7° ± 2.4° and 5.2° ± 0.7° and the energy consumption was 136.4 ± 7.0 Wh. Oxygen plasma etching for 15 min and thermal aging at 120 °C for 24 h (E15H120 24 h) resulted in a static contact and shedding angle of 180.0° ± 0.0° and 1.8° ± 0.2° and energy consumption of 3628.5 ± 82.6 Wh. E15H120 1 h showed a lower shedding angle but had a higher sliding angle of 90°. E15H120 24 h exhibited shedding and sliding angles of less than 10°. Regardless of the thermal aging time, superhydrophobicity was higher in high-density fabrics than in low-density fabrics. The superhydrophobic PP fabric had a similar water vapor transmission rate and air permeability with the untreated PP fabric, and it showed a self-heading property after washing followed by tumble drying and hot pressing.

## 1. Introduction

A superhydrophobic surface, defined as a surface with a static contact angle of 150° or higher and a shedding angle of less than 10° for water [1], exhibits self-cleaning ability for water and stains, and anti-adhesive characteristics against bacteria and microorganisms [2]. Surfaces exhibiting these characteristics can be achieved by imitating the micro–nano double roughness and low surface energy of a lotus leaf [3,4]. These superhydrophobic polymers have properties such as anti-fouling, anti-fogging, anti-freezing, antimicrobial bacteria, and self-cleaning. Applying this surface to clothing materials can improve their convenience by reducing the frequency of washing [5]. Therefore, previous studies have implemented superhydrophobicity by forming a nanoscale surface roughness through the attachment of nanoparticles, electric radiation, plasma etching, or alkali reduction processes on a fabric surface with a unique microroughness [6,7,8,9], followed by coating with fluorine and silane compounds or thermal aging [2,7,10,11]. However, nanoparticles can be absorbed into the human body when they are removed because of low adhesion with the surface; thus, nanoparticles have limited application to clothing materials that are in frequent contact with the body [12,13]. Fluorine compounds are commonly used in surface hydrophobization processes, as they can effectively lower the surface energy [14]. However, they are accumulated in nature owing to their difficult decomposition and can also generate carcinogens, such as perfluorooctanesulfonic (PFOS) and perfluorooctanoic acid (PFOA), once they are decomposed [15]. Due to those human and environmental hazards of fluorine compounds, silicone-based or pyridine-based water repellents are considered as alternatives [16]. However, coating fabrics with hydrophobic water repellents also has limitations, such as reduced breathability, comfort, form, and flexibility [17].

Oh et al. [10,11,18] developed a surface hydrophobization method by thermal aging using hydrophobic materials, such as polyethylene terephthalate (PET), and without using any chemicals. The optimal conditions of thermal aging was set to be 130 °C for 24 h after introducing surface polar groups and nanoroughness using oxygen plasma etching or alkali reduction process. However, PET has a relatively higher glass transition temperature and requires significant time and energy to achieve superhydrophobicity. Additionally, the surface energy of PET (43.1 mN/m [16]) due to the ester bond of –COO, which is the polar group of the molecular chain, is not sufficiently low to replace fluorine- and silicon-based repellents. In contrast, PP used in this study has a lower surface energy of 30.1–32.9 mN/m [19,20,21] and glass transition temperature below room temperature [22]. For the etching gas, we used O_2_ gas to render nanoroughness since it contributes fewer visual changes [23] and has a higher etch rate (nm/min) and average roughness (Ra) [24] than Ar and CF_4_ gas. Hence, an energy-efficient superhydrophobic surface could be achieved with minimal power consumption by thermal aging.

The shedding angle, measured by dropping water droplets on an inclined surface, provides a quantification of superhydrophobicity [25]. However, it cannot reflect the stability of the hydrophobic recovery and wicking effect because of the short contact time between the water droplets and solid surface [25]. In contrast, the sliding angle measures the angle when a water droplet starts to roll down on a surface inclined from 0° to 90° [26]. In this regard, the sliding angle can better evaluate hydrophobic stability; therefore, we used this parameter to compare hydrophobic stability at different thermal aging times.

This study examined the conditions needed to achieve superhydrophobicity on PP fabrics and investigated the differences in the superhydrophobic performance and energy efficiency under different conditions. Oxygen plasma were used to introduce nanoroughness on the PP fabrics, and hydrophobic recovery was induced by thermal aging. The surface and chemical composition changes were analyzed. Superhydrophobicity was evaluated using static contact, shedding, and sliding angles, and self-cleaning ability was tested with silicon carbide particles. In addition, the optimal conditions to minimize energy use while achieving high superhydrophobicity were determined by measuring the energy consumption at different thermal aging temperatures and times. Furthermore, the superhydrophobic behaviors were compared with different weave densities of the fabrics to suggest the optimal conditions to achieve superhydrophobicity for actual clothing production. Finally, the water vapor transmission rate and air permeability were tested for clothing suitability, and the self-healing ability was obtained by a washing test with tumbling drying and hot pressing.

## 2. Materials and Methods 

### 2.1. Materials

For the samples, 100% PP 150D/96F DTY yarns (Raphila, Daegu, Korea) with 72 × 72 plain weave per inch were used. The weave density was changed thrice (Appendix A). A nonionic surfactant (Jayeon, Jeolla-do, Korea) consisting of polyoxyethylenealkyl ether was used as the scouring agent to remove impurities, such as oil stains generated in the filature and weaving process. During oxygen plasma etching, 99.99% oxygen gas was used [11].

### 2.2. Fabrications

The samples (10 × 10 cm^2^) were attached to silicon wafers and inserted into inductively coupled plasma etcher PlasmaLab 80 Plus (Oxford Instrument, Abingdon, UK). The working pressure inside the chamber was maintained at 40 mTorr by injecting oxygen at 20 sccm under vacuum. Plasma etching was performed at a power of 150 W and DC bias of −450 V for 1, 3, 5, 7, 10, and 15 min, respectively. Thermal aging was performed at 40, 60, 80, 100, 110, and 120 °C in a natural convection oven (n-32, DAIHAN Scientific Co., Wonju-si, Korea) for 1, 3, 5, 9, 15, and 24 h. The energy consumption according to the thermal aging temperature and time was measured using a power meter (WT310 Digital Power Meter, Yokogawa, Japan) (Figure 1). The specimen codes were designed as shown in Table 1.

### 2.3. Characterization

#### 2.3.1. Surface and Chemical Composition Analysis

The surface structural changes of the samples were examined using field-emission scanning electron microscopy (FE-SEM, SUPRA/AURIGA, Carl Zeiss, Oberkochen, Germany) for 5000 and 20,000 magnification using High Resolution Digital Image Processing and Analysis System. To impart conductivity, the surface was coated with 20 nm thick platinum using a coating machine (Ion Sputter Coater, G20, GSEM, Suwon, Korea). Atomic force microscopy (AFM, NX-10, Park Systems, Suwon, Korea) was carried out to quantitatively analyze the nanostructures based on plasma etching. The nanoroughness was measured in the section of 3 μm × 3 μm of the specimen. Then, the range of 1 μm × 1 μm was randomly selected and the values of Ra and Rq were calculated by XEI software. Kawabata surface roughness meter (KES-FB4-A Surface Tester, Kato Tech Co., Ltd., Kyoto, Japan) was used to assess the microroughness of the fabric and film based on their weave density. To analyze the changes in the surface components based on plasma etching and thermal aging, the changes in the carbon and oxygen composition up to 10 nm of the surface were analyzed using X-ray photoelectron spectroscopy (AXIS-his, Kratos Analytical, Manchester, UK). The crystallinity and immunoreactivity were measured by X-ray diffractometers (New D8 Advance and D8 Discover, Bruker, Billerica, MA, USA).

#### 2.3.2. Wettability Measurement and Self-Cleaning Test

The goal was to evaluate the surface wettability of the samples, the static contact angle for 3.3 ± 0.2 μL [27] 15 water droplets, and shedding and sliding angles for 12.5 ± 0.2 μL [18]. Water drops were measured using a static contact angle meter (Theta Lite Optical Tensiometer, KSV Instruments, Helsinki, Finland). The self-cleaning test was conducted based on the method proposed by Oh et al. [10]. Specifically, the silicon carbide particles of the superhydrophobic sample were evenly dispersed on a surface inclined at 10°. Self-cleaning property was satisfied when the water droplets dropped 1 cm above the stain rolled for more than 2 cm [18].

#### 2.3.3. Measurement of the Energy Consumption

The energy consumption according to the thermal aging temperature and time was measured using a power meter (WT310 Digital Power Meter, Yokogawa, Tokyo, Japan).

#### 2.3.4. Breathability and Durability Test

To evaluate the breathability of the clothing material, its water vapor transmission rate and air permeability were measured according to KS K 0594:2015 [10] and ASTM D737-04 [28]. In addition, to investigate the washing durability of the sample, a Terg-O-Tometer (Yasuda Seiki, Tokyo, Japan) was used and for self-healing property; it was dried by a 9 kg drum-type heater clothing drier (TROMM RC9D11A, LG Electronics, Seoul, Korea) or heated by hot pressed (WJHP-4550S, DONG-A Machinery Co., Gyeongsang-do, Korea).

## 3. Results

### 3.1. Change in Surface Morphology

With plasma etching performed for an extended time, a nanostructure was clearly formed on the smooth microfiber (Figure 2 and Appendix A). When the plasma etching time was increased to 10 min, the arithmetical mean deviation Ra and root mean square Rq values sharply increased from 30.30 to 124.54 nm and from 37.45 to 147.55 nm, respectively. When the plasma etching time was further increased to 15 min, the Ra and Rq values were 117.82 and 142.45 nm, respectively, showing the slowing increase of the surface roughness. By observing the nanostructure of the three representative samples selected at 5 min intervals using AFM analysis (Appendix A), the nanoroughness value was found to be similar before and after thermal aging, confirming that thermal aging does not influence the surface nanostructure. 

### 3.2. Changes in the Chemical Compositions

The composition analysis result of the untreated PP sample (UT) showed a 0% oxygen ratio since the PP molecular chain was only composed of carbon and hydrogen (Table 2). As the oxygen plasma etching time increased, the surface oxygen ratio increased because of the polar groups attached to the surface. When the E5H120, E10H120, and E15H120 samples were thermally aged for 1 h, the surface oxygen ratio decreased by approximately 50%; however, the sample subjected to plasma etching for 15 min still exhibited a high oxygen ratio. This suggests that a longer oxygen plasma etching time results in more polar functional groups introduced to the surface and a hydrophobic recovery requiring more time [29] Thus, when the E15H120 was thermally aged for 1, 5, and 24 h, the surface chemical composition of UT PP fabric gradually recovered, resulting in hydrophobic recovery. 

### 3.3. Surface Wettability at Various Thermal Aging Temperatures

After plasma etching, the water droplets were immediately absorbed for all samples with a static contact angle of 0° (Figure 3). Surface wettability was observed on the surface of E10H120. The results showed that the water droplets were absorbed within 25 s up to 80 °C, resulting in a static contact angle of 0°. However, at 100 °C, a constant static contact angle of 150° was noted for 10 s, noting a relatively stable hydrophobic recovery. After thermal aging for 24 h at 120 °C, the water droplets exhibited bouncing behavior without attaching to the surface even after repeatedly dropping water droplets (Appendix A). This is attributed to the higher thermal aging temperature that resulted in a higher molecular chain mobility, and quick rearrangement and diffusion of the surface polar groups in the surface [29]. The minimum temperature required for the hydrophobic recovery of the PP film used by Oh et al. [30] was 55 °C, while that in this study was 100 °C since the hydrophobic recovery was determined not only by the glass transition temperature Tg, which indicates molecular mobility, but also by crystallinity and orientation [30]. The PP fabric used in this study and PP film in a previous study [30] has crystallinities of 37.2% and 65.1%, respectively. Therefore, surface hydrophobic recovery at a low temperature would be possible for the PP fabric owing to its lower crystallinity than the PP film. However, the result showed conversely because of the orientations. The PP fabric had a low orientation with unbroken Debye–Scherrer concentric circles [31], whereas the PP film had a high orientation with crescent-shaped broken concentric circles (Appendix A). In this aspect, the orientation had a greater effect on the hydrophobic recovery than crystallinity; hence, the PP film with high degree of orientation is more advantageous for molecular chain mobility [32]. Consequently, the PP fabric used in this study exhibited hydrophobic recovery at a relatively high temperature.

### 3.4. Surface Wettability and Energy Consumption at Various Oxygen Plasma Etching and Thermal Aging Time

To examine the effects of the oxygen plasma etching time, thermal aging temperature, superhydrophobicity, and energy consumption, the static contact angle, shedding angle, and energy consumption were measured at 100, 110, and 120 °C (Figure 4).

For all thermal aging temperatures (100, 110, and 120 °C), the samples exhibited improved superhydrophobicity as the plasma etching time increased to 5, 10, and 15 min (Figure 4a–c). A maximum static contact angle of approximately 120° can be achieved on a smooth PP surface and by lowering the surface energy [31,33,34]. However, depending on the oxygen plasma treatment time and thermal aging, the increase of the etching rate and the recovered surface energy resulted in the reduction of the surface contact area for liquid. As a result, the static contact angles increased while the shedding angles decreased [11]. During thermal aging at 100 °C, longer thermal aging time resulted in a higher contact angle and lower shedding angle (Figure 4a). A similar pattern was observed with thermal aging at 110 and 120 °C. As the temperature increased, the maximum contact angle increased while the shedding angle reached its minimum point (Figure 4a–c). Particularly, there is a significant increase and decrease in the static contact and shedding angles, respectively, during thermal aging for the first 1 h. Superhydrophobicity was noted with 15 min plasma etching, thereby resulting in the maximum contact angle (Figure 4d). At thermal aging at 100 °C for 5 h and 120 °C for 1 h, they had static contact angles of 152.8° ± 2.5° and 162.7° ± 2.4°, shedding angles of 9.6° ± 0.5° and 5.2° ± 0.7°, and energy consumption of 562.6 ± 12.9 Wh and 136.4 ± 7.0 Wh, respectively. In this regard, the E15H120 1 h was found to be advantageous in terms of superhydrophobicity and low energy consumption; thus, it was selected as the most optimal condition for the development of an energy-efficient PP fabric. Furthermore, the E15H120 24 h had the highest superhydrophobicity with a static contact angle of 180.0° ± 0.0° and shedding angle of 1.8° ± 0.2° with an energy consumption of 2672 ± 30 Wh; thus, this was also selected as the best condition to achieve the highest superhydrophobicity. Both samples were used for the succeeding experiments.

### 3.5. Comparison of the Shedding Angle, Sliding Angle, and Self-Cleaning Property

For a detailed comparison of the dynamic behaviors of the water droplets, the shedding and sliding angles were measured. The UT had a static contact, shedding, and sliding angles of 138.7°, 21.0°, and 43.4°, respectively, because of the insufficient nanostructures and hydrophobicity (Figure 5a). In contrast, the UT composing of PP had a higher static contact angle and lower shedding angle than hydrophilic fabrics, such as cotton, owing to the inherent low surface energy and microroughness. E15H120 1 h exhibited high superhydrophobicity with a static contact angle of 162.7° and shedding angle of 5.2°. However, the sliding angle was higher than 90° and the water droplet was pinned to the surface for a long period without roll-off (Figure 5b). In contrast, for the E15H120 24 h, nanostructures were formed on the surface by plasma etching and the surface chemical composition was fully recovered to the level of UT. Thus, superhydrophobicity conditions with a static contact angle of 180.0°, shedding angle of 1.8°, and sliding angle of 9.8° were all satisfied. This difference in the sliding angle according to the thermal aging time can be explained by the lotus and petal effects. Unlike the lotus effect that is satisfied by a combination of sufficient surface roughness and low surface energy, the petal effect appears because of a metastable surface energy or surface roughness [23,35]. Consequently, the E15H120 24 h demonstrated the lotus effect of the Cassie–Baxter state with the complete hydrophobic recovery of UT level. In contrast, the E15H120 1 h showed a petal state with the water droplets pinned to the surface because of incomplete hydrophobic recovery and Cassie–Wenzel transition owing to the remaining polar functional groups on the surface introduced by oxygen plasma etching because of the short thermal aging time (Table 1). When the self-cleaning property was examined, the water droplets did not roll on the UT without a surface structure. In contrast, the E15H120 1 h and E15H120 24 h both exhibited excellent self-cleaning ability with the complete removal of silicon carbides, and the water droplets rolling over 2 cm, because of the surface nanostructures and hydrophobic recovery. This suggests that the self-cleaning property is more affected by the shedding angle than the sliding angle.

### 3.6. Surface Wettability at Various Weave Density

The effects of the microroughness on the static contact, shedding, and sliding angles were compared for the E15H120 1 h and E15H120 24 h with different weave densities (Table 3 and Table 4). First, for the smooth film, the static contact, shedding, and sliding angles were ≥103.8°, ≥90°, and ≥90°, respectively (Appendix A). Meanwhile, the UT with a microstructure exhibited a static contact, shedding, and sliding angles of 143.5°, 17.2° and 31.6°, respectively, at a weave density of 62 × 62 per in. As the weave density of the UT increased to 72 × 72 per in and 82 × 80 per in, the static contact, shedding, and sliding angles decreased to 138.7° and 133.3°, increased to 21.0° and 32.8°, and increased to 43.4° and 57.8°, respectively (Appendix A). This hydrophobic decreasing pattern is attributed to the increased weave density, in which the distance between the fibers are smaller, and the gap and height difference between the microroughness are narrowed, resulting in a smaller roughness factor value and decreased air gap [36]. This is also consistent with the results of the Kawabata surface roughness measurement. As the weave density increased, the surface microroughness value tended to decrease (Appendix A).

By evaluating the surface wettability of E15H120 1 h and E15H120 24 h with different weave densities that were subjected to superhydrophobic processing, higher weave density resulted in a larger static contact angles and smaller shedding angles. Particularly, the E15H120 24 h exhibited increasing static contact angles, decreasing shedding angles, and decreasing sliding angles (Table 2 and Table 3), which is opposite to results for the UT. The increase in weave density increased the nanoroughness per unit area and decreased the contact area with the water droplets, resulting in a larger contact angle and smaller shedding and sliding angles (Appendix A). However, for the E15H120 1 h, the sliding angles still were all greater than 90° regardless of the weave density because of the petal effect of the incomplete hydrophobic recovery. This suggests that fabrics with surface nanostructures have better static contact, shedding, and sliding angles at a higher density when complete hydrophobic recovery occurs by thermal aging.

### 3.7. Breathability and Durability Test

The water vapor transmission rate (WVTR) and air permeability were measured to evaluate the suitability of the fabric as a clothing material (Figure 6a). The water vapor transmission rate is affected by the physical porosity and surface hydrophilicity. The E15 sample exhibited a higher WVTR than the UT since the polar functional groups introduced on the surface facilitated water absorption during plasma etching. In addition, the E15H120 1 h had a lower water vapor transmission rate than the E15, which was similar to that of the UT. This is because of the rearrangement of the surface polar groups introduced by plasma etching during thermal aging, thereby hydrophobizing the surface again [11]. For air permeability, the E15 remained approximately constant compared to the UT, whereas the E15H120 1 h exhibited a slight decrease. This is because of the effect of the physical density or porosity of the fabric to air permeability; the air permeability decreased because of the thermal contraction by thermal aging, which decreased the gaps and voids in the fabric [10,16]. However, the plasma etching at a nanoscale level on the surface did not change the properties of the entire bulk of the material, resulting in minimal effects to the air permeability and showing similar level of air permeability with the UT [8,11].

The durability of the developed fabrics in actual clothing was evaluated using washing with water with a heater-type clothing dryer and hot press that simulated ironing (Figure 6b,c). The E15H120 1 h and E15H120 24 h exhibited high contact angles before washing that decreased to 128.5° and 136.9°, respectively, after washing. This can be attributed to the nanoroughness damaged by washing and rearranging the internal polar groups, including oxygen, on the surface when in contact with water [10,11,18]. When the samples were heat-treated in a heater-type clothing dryer, their contact angles increased to 146.0° and 147.0°, respectively, and further increased to 155.7° and 159.4°, respectively, under a hot press, indicating their self-healing properties. This is because of the re-application of thermal aging that rearranged the polar functional groups in the surface and generated the self-healing property, indicating the management durability [10,11,18].

## 4. Conclusions

This study used PP fabric, which has an advantage for hydrophobic recovery owing to intrinsic low surface energy and low glass transition temperature, and the newly developed fabric using oxygen plasma etching and short-term thermal aging in human-friendly, energy-efficient superhydrophobic samples. The superhydrophobicity and energy consumption of the samples were measured after thermal aging at 100, 110, and 120 °C for different plasma etching times (5, 10, and 15 min). The results showed that a longer the plasma etching time resulted in a higher superhydrophobicity, while higher thermal aging temperature resulted in a faster hydrophobic recovery. Among the conditions to achieve superhydrophobicity, thermal aging for 1 h at 120 °C after plasma etching for 15 min (E15H120 1 h) demonstrated the lowest energy consumption; thus, it was selected as the optimal condition to develop a human-friendly, energy-efficient superhydrophobic PP fabric. Meanwhile, thermal aging for 24 h at 120 °C after oxygen plasma etching for 15 min (E15H120 24 h) exhibited the highest superhydrophobicity with a contact angle of 180° and shedding angle of 1.8° ± 0.2°; hence, it was selected as the condition for maximum superhydrophobicity. Furthermore, both of the two samples, E15H120 1 h and E15H120 24 h, showed excellent self-cleaning properties. In terms of dynamic wetting behavior, however, they were different. E15H120 1 h displayed a petal state with a sliding angle greater than 90° and pinning of water droplets, but E15H120 24 h exhibited a lotus state with a sliding angle of 9.8°. When the microroughness was adjusted by changing the weave density, the high-density fabric (82 × 80 per in) proved to be more favorable for superhydrophobicity than the low-density fabric (62 × 62 per in). Moreover, the water vapor transmission rate and air permeability of the treated samples were similar with the original. In terms of the washing durability, the static contact angle decreased after washing but increased with additional thermal aging, tumble drying, or hot plate, indicating the management durability owing to the self-healing property. Therefore, this study developed an eco-friendly recyclable superhydrophobic PP fabric that can minimize energy consumption through thermal aging for a short time without using additional compounds in the hydrophobization process. 

## Figures and Tables

**Figure 1 polymers-12-02756-f001:**
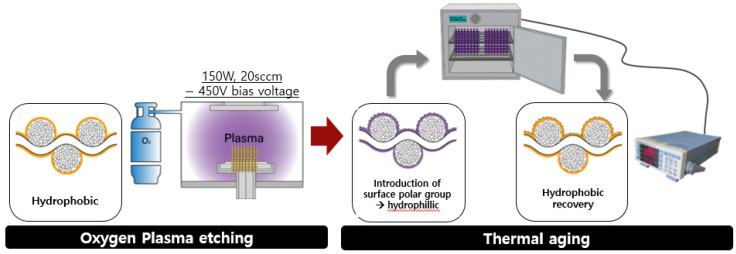
Schematic of the plasma etching and thermal aging process to develop a superhydrophobic PP fabric.

**Figure 2 polymers-12-02756-f002:**
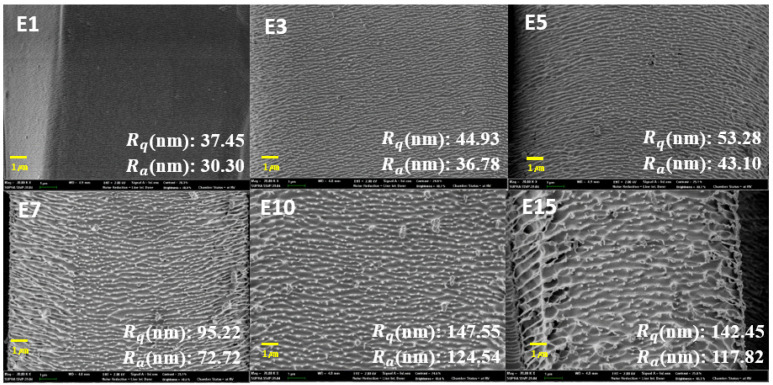
Results of scanning electron microscopy and atomic force microscopy of the plasma-etched specimens for 1, 3, 5, 7, 10, and 15 min, and their arithmetical mean deviation Ra and root mean squared Rq (magnification of × 20,000, scale bar: 1 μm).

**Figure 3 polymers-12-02756-f003:**
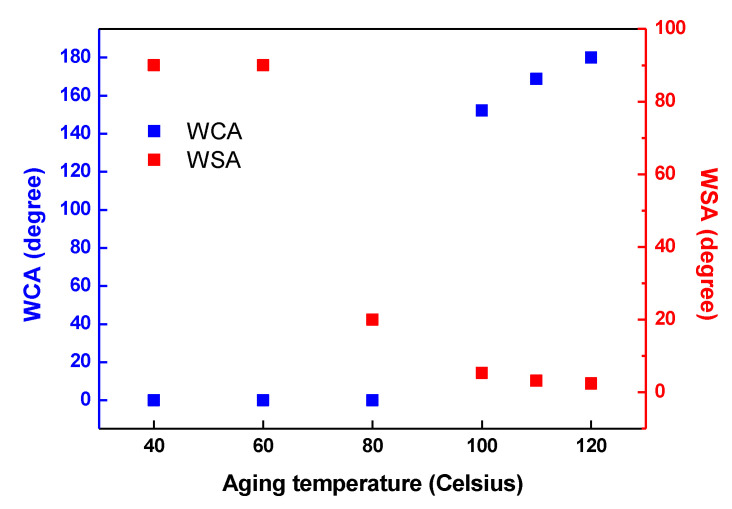
Water contact angle and water shedding angle of the water droplets on the samples that underwent plasma etching for 10 min and thermal aging for 24 h depending at various aging temperatures.

**Figure 4 polymers-12-02756-f004:**
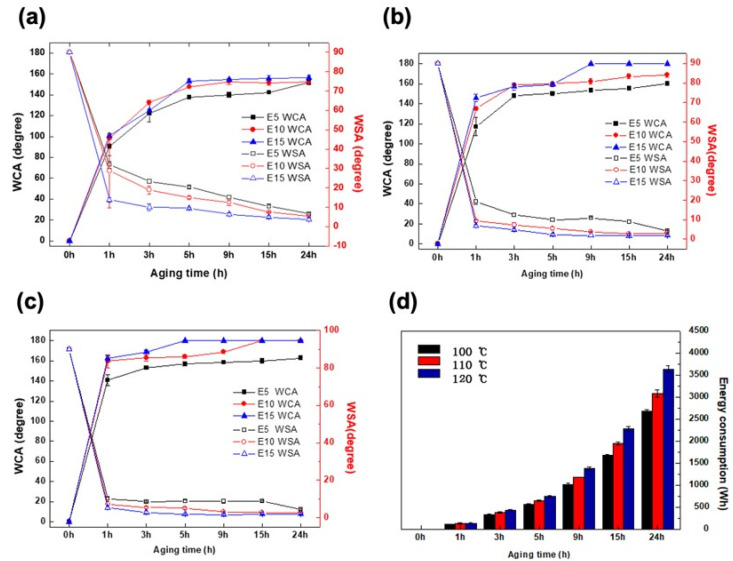
Static contact and shedding angles of the PP fabrics that underwent plasma etching for 5, 10, and 15 min and thermal aging at 100 °C (**a**), 110 °C (**b**), and 120 °C (**c**). Energy consumption at the thermal aging temperatures of 100, 110, and 120 °C and times of 1, 3, 5, 9, 15, and 24 h (**d**).

**Figure 5 polymers-12-02756-f005:**
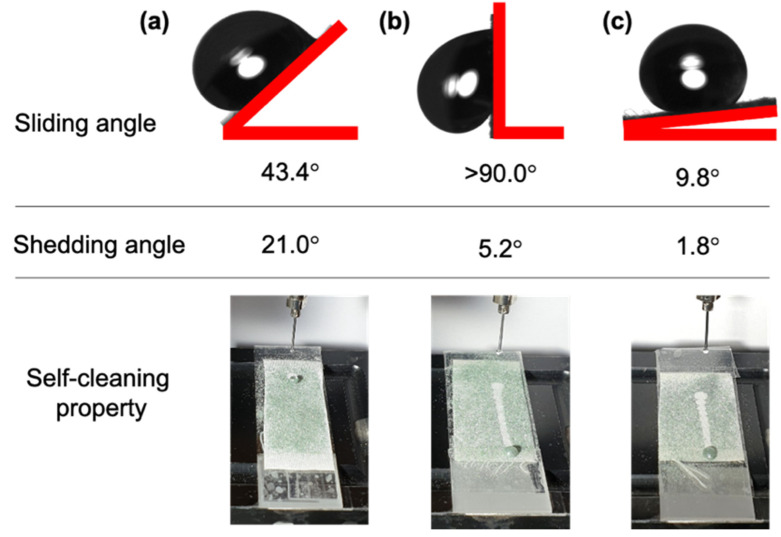
Sliding angles, shedding angles, and self-cleaning property of the untreated PP fabric (UT) (**a**) and superhydrophobic specimen treated with plasma etching for 15 min and thermal aging at 120 °C for 1 h (E15H120 1 h) (**b**), and plasma etching for 15 min and thermal aging at 120 °C for 24 h (E15H120 24 h) (**c**).

**Figure 6 polymers-12-02756-f006:**
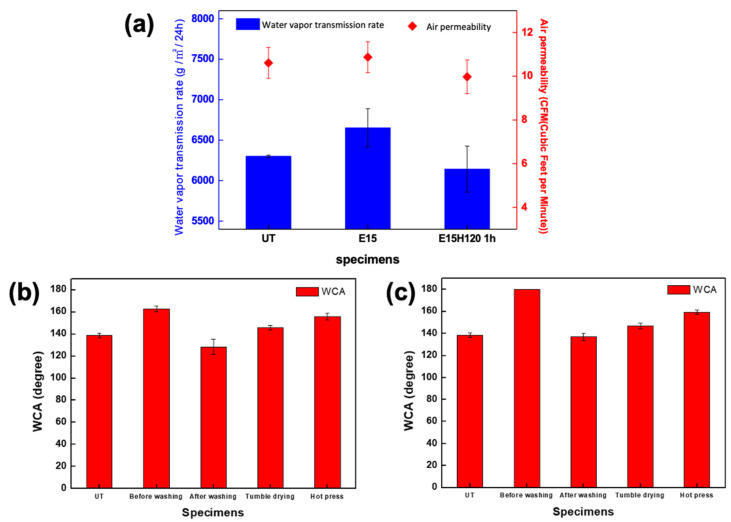
Water vapor transmission rate and air permeability of untreated PP fabric (UT), PP fabric treated with plasma etching for 15 min (E15), and PP fabric treated with plasma etching for 15 min and thermal aging at 120 °C for 1 h (E15H120 1 h) (**a**). Static contact angles before and after washing, tumble drying, and hot-pressing of E15H120 1 h (**b**) and the PP fabric treated with plasma etching for 15 min and thermal aging at 120 °C for 24 h (E15H120 24 h) (**c**).

**Table 1 polymers-12-02756-t001:** Sample codes during the treatments.

Sample Codes																					
**Etching time** **(min)**	E1	E3	E5						E7	E10						E15					
**Thermal aging Temperature (°C)**	H120	H120	H100		H110		H120		H120	H40	H60	H80	H100	H110	H120	H100		H110		H120	
**Thermal aging time (h)**	24	24	1	3	5	9	15	24	24	1	3	5	9	15	24	1	3	5	9	15	24

**Table 2 polymers-12-02756-t002:** Atomic composition of the plasma etching and thermal aging treatment of the specimen for various durations.

Specimen Codes		Atomic Composition (at.%)		
		C	O	Total
**UT**		100	0	100
**E5**		77.80	22.20	100
**E10**		73.52	26.48	100
**E15**		69.97	30.03	100
**E5H120 1 h**		95.52	4.48	100
**E10H120 1 h**		84.48	15.52	100
**E15120**	**1 h**	78.50	21.50	100
	**5 h**	84.48	15.52	100
	**24 h**	100	0	100

**Table 3 polymers-12-02756-t003:** Water contact, shedding, and sliding angles of the polypropylene (PP) fabric that underwent plasma etching for 15 min and thermal aging for 1 h (E15H120 1 h).

	Weave Density (per in)		
	62 × 62	72 × 72	82 × 80
**WCA (°)**	155.2 ± 1.6	162.7 ± 2.4	166.3 ± 6.2
**ShA (°)**	8.4 ± 1.0	5.2 ± 0.7	2.2 ± 0.7
**SA (°)**	>90	>90	>90

**Table 4 polymers-12-02756-t004:** Water contact, shedding, and sliding angles of the PP fabric that underwent plasma etching for 15 min and thermal aging for 24 h (E15H120 24 h).

	Weave Density (per in)		
	62 × 62	62 × 62	62 × 62
**WCA (°)**	179.3 ± 2.3	180 ± 0.0	180 ± 0.0
**ShA (°)**	2.8 ± 0.4	1.8 ± 0.2	1.6 ± 0.2
**SA (°)**	11.2 ± 2.9	9.8 ± 2.3	4.2 ± 0.7

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
