# Peer review of "Development of Energy-Efficient Superhydrophobic Polypropylene Fabric by Oxygen Plasma Etching and Thermal Aging"

_polymers, 2020, doi:10.3390/polym12112756_

Round 1

Reviewer 1 Report

This paper reports on oxygen plasma etching of polymer materials.

Water protection may be one of the important role for polymer materials, so this surface modification method may be useful for practical propose. Therefore, it should be published in this journal essentially.

Reliable experiments and readable description were high-level.

Before acceptance, only some aspects of concept should be improved.

(1) Role of surface shape.

Is it important surface shape (somewhat damaged by oxygen plasma)?

(2) Possibility of chemical reactions.

 Is there possibility of chemical reactions for terminal groups of polymers by oxygen?

(3) Superhydrophobic solvents

Does such modified polymers exhibit solvent dependency about superhydrophobic features?

That's all.

Author Response

We highlighted the revised sentence in the manuscript. Please see the attachment which is the revised manuscript.

Reviewer 1.

This paper reports on oxygen plasma etching of polymer materials.

Water protection may be one of the important roles for polymer materials, so this surface modification method may be useful for practical propose. Therefore, it should be published in this journal essentially.

Reliable experiments and readable descriptions were high-level.

Before acceptance, only some aspects of the concept should be improved.

(1) Role of surface shape.

Is it an important surface shape (somewhat damaged by oxygen plasma)?

 - The nanostructured surface made plays a high role in exhibiting superhydrophobicity as reported in various studies. The hairy structures make the surface have a lower contact area with a water droplet and therefore, due to the diminished attached area and low surface energy, the water droplet can be a sphere shape, showing superhydrophobicity. And as the plasma etching is longer, the surface etching was deeper by the etching time of 15 min, while maintaining the physical properties.

(2) Possibility of chemical reactions.

 Is there a possibility of chemical reactions for terminal groups of polymers by oxygen?

 - There are chemical reactions on the surface of PP fabric during the oxygen plasma etching. During the process, C-C bonds on the fabric are broken by the activated oxygens; radical oxygen, and the products in the form of CO2 and O2. Therefore, the nanostructures which have carbon radicals on the surface are made. Consequently, the reaction between carbon radicals and oxygen or hydrogen in the air rapidly occurs and the induced polar functional groups having the forms of -OH are created as seen in Table 2.

(3) Superhydrophobic solvents

Does such modified polymers exhibit solvent dependency about superhydrophobic features

  • The plasma etched PP fabric will at least exhibit oleophobicity due to the reduced adhesive energy.
    In addition, referring to the previous research reporting that the plasma etched and thermal aged PET surface showed the static contact angles higher than 160° against the liquid with the surface tension of 42 dyne/cm, we highly assume that the developed PP fabric has oleophobicity.

Reviewer 2 Report

Dear Authors,

I have read the text and some questions raised. Enlisted please find my comments.

Overall. General English grammar revision (minor spelling errors).

Introduction. Authors stated “Surfaces exhibiting these characteristics can be achieved by imitating the micro–nano double roughness and low surface energy of a lotus leaf”. Please add a brief introduction about various uses of polymeric nanomaterials.

Introduction. Authors stated “Applying this surface to clothing materials can improve their convenience by reducing the frequency of washing”. Please add a reference for this statement.

Introduction. Authors stated “Fluorine compounds are commonly used in a surface hydrophobization process as they can effectively lower the surface energy”. Please add a reference for this statement.

Introduction. Authors stated “The shedding angle, measured by dropping water droplets on an inclined surface, provides a quantification of superhydrophobicity”. Please add a reference for this statement.

Materials and Methods. Authors stated ” For the samples, 100% PP 150D/96F DTY yarns (Raphila, Korea) with 72 × 72 plain weave per inch were used”. Please add City of the Manufacturer.

Materials and Methods. Authors stated “A nonionic surfactant provided by a company was used as the scouring agent to remove impurities, such as oil stains generated in the filature and weaving process”. Please add commercial name, along with manufacturer details.

Materials and Methods. Authors stated “During oxygen plasma etching, 99.99% oxygen gas was used”. Please add a reference for this method.

Materials and Methods. Authors stated ”The samples (10 × 10 cm) were attached to silicon wafers and inserted into inductively coupled plasma etcher PlasmaLab 80 Plus (Oxford Instrument, UK)”. Please add City of the Manufacturer.

Materials and Methods. Authors stated “Thermal aging was performed at 40, 60, 80, 100, 110, and 120 °C in a natural convection oven”. Please add a reference for this method.

Materials and Methods. Authors used a ”convection oven (n-32, 93 DAIHAN Scientific Co., Korea)”. Please add City of the Manufacturer.

Materials and Methods. Authors stated “The surface structural changes of the samples were examined using field-emission scanning electron microscopy”. Please add magnification and Software used.

Materials and Methods. Authors stated “The surface structural changes of the samples were examined using field-emission scanning electron microscopy (FE-SEM, SUPRA/AURIGA, Carl Zeiss, Germany) … coating machine (Ion Sputter Coater, G20, GSEM, Korea) … Atomic force microscopy (AFM, NX-10, Park Systems, Korea) … Kawabata surface roughness 107 meter (KES-FB4-A Surface Tester, Kato Tech Co., Ltd., Japan) … X-ray photoelectron spectroscopy (AXIS-his, Kratos Analytical, UK) … X-ray diffractometers 112 (New D8 Advance and D8 Discover, Bruker, USA) … static 116 contact angle meter (Theta Lite Optical Tensiometer, KSV Instruments, Finland) … power meter (WT310 Digital Power Meter, Yokogawa, Japan) … Terg-O-Tometer (Yasuda Seiki, Japan) … 9 kg drum-type heater clothing drier (TROMM RC9D11A, LG Electronics, Korea) … (WJHP-4550S, DONG-A Machinery Co., Korea)”. Please add Cities of all the Manufacturers.

Materials and Methods. Authors stated “To evaluate the surface wettability of the samples, the static contact angle for 3.3 ± 0.2 μL water 115 droplets, and shedding and sliding angles for 12.5 ± 0.2 μL water drops were measured”. Please add a reference for this method.

Materials and Methods. Authors stated “Self-cleaning property was satisfied when the water droplets dropped 1 cm above the stain rolled for more than 2 cm”. Please add a reference for this statement.

Materials and Methods. Authors stated “To evaluate the breathability of the clothing material, its water vapor transmission rate and air permeability were measured according to KS K 0594:2015 and ASTM D737-04”. Please add a reference for these standards.

Materials and Methods. Was data submitted to statistical analysis? If so please name the tests used and along with Software name.

Figure 2. Please add magnification in the Figure caption.

Results. Authors stated “When the plasma etching time was increased to 10 min, the arithmetical mean deviation Ra and root mean square Rq values sharply increased from 30.30 to 124.54 nm and from 37.45 to 147.55 nm, respectively”. In Materials and Methods section it should be specified the number of specimens tested for each group and the Software used to calculate the mean values.

Results and Discussion. Please add a paragraph concerning the limitations of the present report.

Results and Discussion. Many variables could alter the results of the present report. Please point out that “Thermal aging has been tested in the present report. However, many factors could alter surface characteristics and mechanical behavior of polymeric nanomaterials, such as the presence of nanofibers (Preparation and Characterization of Soy Protein Isolate-Based Nanocomposite Films with Cellulose Nanofibers and Nano-Silica via Silane Grafting. Qin Z, Mo L, Liao M, He H, Sun J. Polymers (Basel). 2019 Nov 7;11(11):1835.) particle size (A Comparative Study of Particle Size Distribution of Graphene Nanosheets Synthesized by an Ultrasound-Assisted Method. Amaro-Gahete J, Benítez A, Otero R, Esquivel D, Jiménez-Sanchidrián C, Morales J, Caballero Á, Romero-Salguero FJ. Nanomaterials (Basel). 2019 Jan 26;9(2):152.) and wear (Effect of Long-Term Brushing on Deflection, Maximum Load, and Wear of Stainless Steel Wires and Conventional and Spot Bonded Fiber-Reinforced Composites. Scribante A, Vallittu P, Lassila LVJ, Viola A, Tessera P, Gandini P, Sfondrini MF. Int J Mol Sci. 2019 Nov 30;20(23):6043.). Therefore further studies are needed in order to take into careful account also these variables”.

Figure 4. Please enlarge a bit the graphs in order to increase readability.

Figure 5. Please enlarge a bit the figure in order to increase readability.

Figure 6. Please add statistical significance among various groups.

Conclusions. This section could be shortened.

References. Some studies are quite old (1997; 2000; 1999). Where possible, please switch with some more modern research.

Author Response

We highlighted the revised sentence in the manuscript. Please find the attached file which is the revised manuscript. 

Reviewer 2.

Dear Authors,

I have read the text and some questions raised. Enlisted please find my comments.

Overall. General English grammar revision (minor spelling errors).

Introduction. Authors stated “Surfaces exhibiting these characteristics can be achieved by imitating the micro–nano double roughness and low surface energy of a lotus leaf”. Please add a brief introduction about various uses of polymeric nanomaterials.

We added the short introduction about diverse applications of superhydrophobic materials as below;

Page 1,

These superhydrophobic polymers have properties such as anti-fouling, anti-fogging, anti-freezing, antimicrobial bacteria and self-cleaning.

Introduction. Authors stated “Applying this surface to clothing materials can improve their convenience by reducing the frequency of washing”. Please add a reference for this statement.

We would like to add a reference after the sentence;

Applying this surface to clothing materials can improve their convenience by reducing the frequency of washing [5].

  1. Dalawai, S.P.; Saad Aly, M.A.; Latthe, S.S.; Xing, R.; Sutar, R.S.; Nagappan, S.; Ha, C.-S.; Kumar Sadasivuni, K.; Liu, S. Recent Advances in durability of superhydrophobic self-cleaning technology: A critical review. Progress in Organic Coatings 2020, 138, 105381, doi:10.1016/j.porgcoat.2019.105381.

Introduction. Authors stated “Fluorine compounds are commonly used in a surface hydrophobization process as they can effectively lower the surface energy”. Please add a reference for this statement.

We added a reference in order to support the statement;

Page 1-2,

Fluorine compounds are commonly used in a surface hydrophobization process as they can effectively lower the surface energy [14]

  1. Privett, B.J.; Youn, J.; Hong, S.A.; Lee, J.; Han, J.; Shin, J.H.; Schoenfisch, M.H. Antibacterial Fluorinated Silica Colloid Superhydrophobic Surfaces. Langmuir 2011, 27, 9597–9601, doi:10.1021/la201801e.

Introduction. Authors stated “The shedding angle, measured by dropping water droplets on an inclined surface, provides a quantification of superhydrophobicity”. Please add a reference for this statement.

We used the previous reference which is the number of 25 and added the reference at the end of the sentence;

Page 4,

The shedding angle, measured by dropping water droplets on an inclined surface, provides a quantification of superhydrophobicity [25].

  1. Zimmermann, J.; Seeger, S.; Reifler, F.A. Water shedding angle: a new technique to evaluate the water-repellent properties of superhydrophobic surfaces. Textile Research Journal 2009, 79, 1565–1570.

Materials and Methods. Authors stated ” For the samples, 100% PP 150D/96F DTY yarns (Raphila, Korea) with 72 × 72 plain weave per inch were used”. Please add City of the Manufacturer.

We add city of the manufacturer;

Page 2,

100% PP 150D/96F DTY yarns (Raphila, Daegu, Korea)

Materials and Methods. Authors stated “A nonionic surfactant was used as the scouring agent to remove impurities, such as oil stains generated in the filature and weaving process”. Please add commercial name, along with manufacturer details.

We revised the sentence as below;

Page 2,

A nonionic surfactant(Jayeon,Jeolla-do,Korea)  consisting of an polyoxyethylenealkyl etherwas used as the scouring agent to remove impurities, such as oil stains generated in the filature and weaving process.

Materials and Methods. Authors stated “During oxygen plasma etching, 99.99% oxygen gas was used”. Please add a reference for this method.

We added a reference of validating the sentence of the sentence;

Page 2,

During oxygen plasma etching, 99.99% oxygen gas was used.[11]

Materials and Methods. Authors stated ”The samples (10 × 10 cm) were attached to silicon wafers and inserted into inductively coupled plasma etcher PlasmaLab 80 Plus (Oxford Instrument, UK)”. Please add City of the Manufacturer.

We revised the sentence as below;

Page 2,

The samples (10 × 10 cm) were attached to silicon wafers and inserted into inductively coupled plasma etcher PlasmaLab 80 Plus (Oxford Instrument, Abingdon,UK)

Materials and Methods. Authors stated “Thermal aging was performed at 40, 60, 80, 100, 110, and 120 °C in a natural convection oven”. Please add a reference for this method.

At first, the thermal aging temperature was set below the melting point of polypropylene; 160°C. Below the melting point, the critical temperature showing the hydrophobic recovery was found by changing the temperature every 20° C from the temperature of 40 ° C. Therefore, based on SI Figure 4, the starting temperature showing hydrophobic recovery was 100°C and this temperature was regarded as critical temperature. Then, the hydrophobic recovery was compared by modifying the temperature every 10 °C from 100 °C to 120 °C.

Materials and Methods. Authors used a ”convection oven (n-32, 93 DAIHAN Scientific Co., Korea)”. Please add City of the Manufacturer.

We modified the sentence as below;

Page 3,

convection oven (n-32, 93 DAIHAN Scientific Co.,Wonju-si, Korea)

Materials and Methods. Authors stated “The surface structural changes of the samples were examined using field-emission scanning electron microscopy”. Please add magnification and Software used.

We changed the sentence and added the magnification and the name of software as below;

Page 3,

The surface structural changes of the samples were examined using field-emission scanning electron microscopy(FE-SEM, SUPRA/AURIGA, Carl Zeiss, Oberkochen, Germany) for 5,000 and 20,000 magnification using High Resolution Digital Image Processing and Analysis System.

Materials and Methods. Authors stated “The surface structural changes of the samples were examined using field-emission scanning electron microscopy (FE-SEM, SUPRA/AURIGA, Carl Zeiss, Germany) … coating machine (Ion Sputter Coater, G20, GSEM, Korea) … Atomic force microscopy (AFM, NX-10, Park Systems, Korea) … Kawabata surface roughness 107 meter (KES-FB4-A Surface Tester, Kato Tech Co., Ltd., Japan) … X-ray photoelectron spectroscopy (AXIS-his, Kratos Analytical, UK) … X-ray diffractometers 112 (New D8 Advance and D8 Discover, Bruker, USA) … static 116 contact angle meter (Theta Lite Optical Tensiometer, KSV Instruments, Finland) … power meter (WT310 Digital Power Meter, Yokogawa, Japan) … Terg-O-Tometer (Yasuda Seiki, Japan) … 9 kg drum-type heater clothing drier (TROMM RC9D11A, LG Electronics, Korea) … (WJHP-4550S, DONG-A Machinery Co., Korea)”. Please add Cities of all the Manufacturers.

We added all cities of all the equipment as below;

Page 3-4,

coating machine (Ion Sputter Coater, G20, GSEM,Suwon, Korea)

Atomic force microscopy (AFM, NX-10, Park Systems, Suwon, Korea)

Kawabata surface roughness 107 meter (KES-FB4-A Surface Tester, Kato Tech Co., Ltd.,nKyoto, Japan)

X-ray photoelectron spectroscopy (AXIS-his, Kratos Analytical, manchester, UK)

X-ray diffractometers 112 (New D8 Advance and D8 Discover, Bruker, Billerica, Massachusetts, USA)

static 116 contact angle meter (Theta Lite Optical Tensiometer, KSV Instruments, Helsinki, Finland)

power meter (WT310 Digital Power Meter, Yokogawa,Tokyo, Japan)
Terg-O-Tometer (Yasuda Seiki, Tokyo, Japan)
9 kg drum-type heater clothing drier (TROMM RC9D11A, LG Electronics,Seoul, Korea

(WJHP-4550S, DONG-A Machinery Co., Gyeongsang-do, Korea).

Materials and Methods. Authors stated “To evaluate the surface wettability of the samples, the static contact angle for 3.3 ± 0.2 μL water 115 droplets, and shedding and sliding angles for 12.5 ± 0.2 μL water drops were measured”. Please add a reference for this method.

We added a reference as below;

Page 4,

To evaluate the surface wettability of the samples, the static contact angle for 3.3 ± 0.2 μL [27] 15 water droplets, and shedding and sliding angles for 12.5 ± 0.2 μL [18].

  1. Oh, J.; Park, C.H. Colorful Fluorine‐Free Superhydrophobic Polyester Fabric Prepared via Disperse Dyeing Process. Adv. Mater. Interfaces 2020, 7, 2000127, doi:10.1002/admi.202000127.
  2. Wei, Q.; Schlaich, C.; Prévost, S.; Schulz, A.; Böttcher, C.; Gradzielski, M.; Qi, Z.; Haag, R.; Schalley, C.A. Supramolecular Polymers as Surface Coatings: Rapid Fabrication of Healable Superhydrophobic and Slippery Surfaces. Adv. Mater. 2014, 26, 7358–7364, doi:10.1002/adma.201401366.

Materials and Methods. Authors stated “Self-cleaning property was satisfied when the water droplets dropped 1 cm above the stain rolled for more than 2 cm”. Please add a reference for this statement.

We added a reference for the sentence as below;

Page 4,

Self-cleaning property was satisfied when the water droplets dropped 1 cm above the stain rolled for more than 2cm [18].

Materials and Methods. Authors stated “To evaluate the breathability of the clothing material, its water vapor transmission rate and air permeability were measured according to KS K 0594:2015 and ASTM D737-04”. Please add a reference for these standards.

We would like to add references for the method as below;

Page 4,

KS K 0594:2015 [10] and ASTM D737-04 [28].

  1. Oh, J.; Park, C.H. Robust Fluorine‐Free Superhydrophobic PET Fabric Using Alkaline Hydrolysis and Thermal Hydrophobic Aging Process. Macromolecular Materials and Engineering 2018, 303, 1700673.
  2. American Society for Testing and Materials (ASTM) Standard Test Method for Air Permeability of Textile Fabrics.(ASTM D737-04).; ASTM Philadelphia, PA, 2012.

Materials and Methods. Was data submitted to statistical analysis? If so please name the tests used and along with Software name.

We did not submit the statical analysis.

Figure 2. Please add magnification in the Figure caption.

Page 4-5;

We already added the magnification in the Figure 2 caption; Figure 2. Results of scanning electron microscopy and atomic force microscopy of the plasma-etched specimens for 1, 3, 5, 7, 10, and 15 min, and their arithmetical mean deviation Ra and root mean squared Rq (magnification of × 20,000, scale bar: 1 μm)

Results. Authors stated “When the plasma etching time was increased to 10 min, the arithmetical mean deviation Ra and root mean square Rq values sharply increased from 30.30 to 124.54 nm and from 37.45 to 147.55 nm, respectively”. In Materials and Methods section it should be specified the number of specimens tested for each group and the Software used to calculate the mean values.

We added the below sentence in the materials and methods;

Page 3,

The nanoroughness was measured in the section of 3 3  of the specimen. Then, the range of 1  1  was randomly selected and the values of Ra and Rq were calculated by XEI software.

Results and Discussion. Please add a paragraph concerning the limitations of the present report.

Results and Discussion. Many variables could alter the results of the present report. Please point out that “Thermal aging has been tested in the present report. However, many factors could alter surface characteristics and mechanical behavior of polymeric nanomaterials, such as the presence of nanofibers (Preparation and Characterization of Soy Protein Isolate-Based Nanocomposite Films with Cellulose Nanofibers and Nano-Silica via Silane Grafting. Qin Z, Mo L, Liao M, He H, Sun J. Polymers (Basel). 2019 Nov 7;11(11):1835.) particle size (A Comparative Study of Particle Size Distribution of Graphene Nanosheets Synthesized by an Ultrasound-Assisted Method. Amaro-Gahete J, Benítez A, Otero R, Esquivel D, Jiménez-Sanchidrián C, Morales J, Caballero Á, Romero-Salguero FJ. Nanomaterials (Basel). 2019 Jan 26;9(2):152.) and wear (Effect of Long-Term Brushing on Deflection, Maximum Load, and Wear of Stainless Steel Wires and Conventional and Spot Bonded Fiber-Reinforced Composites. Scribante A, Vallittu P, Lassila LVJ, Viola A, Tessera P, Gandini P, Sfondrini MF. Int J Mol Sci. 2019 Nov 30;20(23):6043.). Therefore, further studies are needed in order to take into careful account also these variables”.

Authors thank for suggesting those good insights. Considering the factors you mentioned, we will proceed with further studies.

Figure 4. Please enlarge a bit the graphs in order to increase readability.

We changed the size of the graphs to enhance readability for readers.

Figure 5. Please enlarge a bit the figure in order to increase readability.

We enlarged the size of the Figure 5.

Page 8;

Figure 6. Please add statistical significance among various groups.

We did not conduct on researching statistical significance.

Conclusions. This section could be shortened.

We changed the conclusion to be shortened and see the below;

Page 10;

This study used PP fabric which has an advantage for hydrophobic recovery owing to intrinsic   low surface energy and low glass transition temperature and the fabric newly developed using oxygen plasma etching and short-term thermal aging is human-friendly, energy-efficient superhydrophobic. The superhydrophobicity and energy consumption of the samples were measured after thermal aging at 100, 110, and 120 °C for different plasma etching times (5, 10, and 15 min). The results showed that a longer the plasma etching time resulted in a higher superhydrophobicity, while higher thermal aging temperature resulted in a faster hydrophobic recovery. Among the conditions to achieve superhydrophobicity, thermal aging for 1 h at 120 °C after plasma etching for 15 min (E15H120 1h) demonstrated the lowest energy consumption; thus, it was selected as the optimal condition to develop a human-friendly, energy-efficient superhydrophobic PP fabric. Meanwhile, thermal aging for 24 h at 120 °C after oxygen plasma etching for 15 min (E15H120 24h) exhibited the highest superhydrophobicity with a contact angle of 180° and shedding angle of 1.8° ± 0.2°; hence, it was selected as the condition for maximum superhydrophobicity. Furthermore, both of the two samples, E15H120 1h and E15H120 24h showed excellent self-cleaning properties. In the dynamic wetting behavior, however, they showed difference. E15H120 1h displayed a petal state with a sliding angle greater than 90° and pinning of water droplets, but E15H120 24h exhibited a lotus state with a sliding angle of 9.8°. When the microroughness was adjusted by changing the weave density, the high-density fabric (82 × 80 per in) proved to be more favorable for superhydrophobicity than the low-density fabric (62 × 62 per in). Moreover, the water vapor transmission rate and air permeability of the treated samples were similar with the original. In terms of the washing durability, the static contact angle decreased after washing, but increased with additional thermal aging; tumble drying or hot plate, indicating the management durability owing to the self-healing property. Therefore, this study developed an eco-friendly recyclable superhydrophobic PP fabric that can minimize energy consumption through thermal aging for a short time without using additional compounds in the hydrophobization process.

References. Some studies are quite old (1997; 2000; 1999). Where possible, please switch with some more modern research.

We changed the old references to the new ones as below;

Page 11,

  1. Liu, Y.; Xin, J.H.; Choi, C.-H. Cotton Fabrics with Single-Faced Superhydrophobicity. Langmuir 2012, 28, 17426–17434, doi:10.1021/la303714h.
  2. Jung, Y.C.; Bhushan, B. Contact angle, adhesion and friction properties of micro-and nanopatterned polymers for superhydrophobicity. Nanotechnology 2006, 17, 4970–4980, doi:10.1088/0957-4484/17/19/033.
  3. Erbil, H.Y. ;l. &i. ;r&i. ;m Transformation of a Simple Plastic into a Superhydrophobic Surface. Science 2003, 299, 1377–1380, doi:10.1126/science.1078365

Round 2

Reviewer 2 Report

Authors revised the text